https://doi.org/10.1038/s41467-019-10284-z | OPEN

# Projected losses of global mammal and bird ecological strategies

Robert S.C. Cooke[1,2,3], Felix Eigenbrod [1,2] & Amanda E. Bates [4,5]

Species, and their ecological strategies, are disappearing. Here we use species traits to quantify the current and projected future ecological strategy diversity for 15,484 land mammals and birds. We reveal an ecological strategy surface, structured by life-history (fast–slow) and body mass (small–large) as one major axis, and diet (invertivore–herbivore) and habitat breadth (generalist–specialist) as the other. We also find that of all possible trait combinations, only 9% are currently realized. Based on species' extinction probabilities, we predict this limited set of viable strategies will shrink further over the next 100 years, shifting the mammal and bird species pool towards small, fast-lived, highly fecund, insect-eating, generalists. In fact, our results show that this projected decline in ecological strategy diversity is much greater than if species were simply lost at random. Thus, halting the disproportionate loss of ecological strategies associated with highly threatened animals represents a key challenge for conservation.

[1] Biological Sciences, University of Southampton, Southampton SO17 1BJ, UK. [2] Geography and Environment, University of Southampton, Southampton SO17 1BJ, UK. [3] Marwell Wildlife, Thompson's Lane, Colden Common, Winchester SO21 1JH, UK. [4] Department of Ocean Sciences, Memorial University of Newfoundland, St. John's, NL A1C 5S7, Canada. [5] Ocean and Earth Science, National Oceanography Centre, Southampton, University of Southampton, Southampton SO14 3ZH, UK. Correspondence and requests for materials should be addressed to R.S.C.C. (email: R.S.Cooke@soton.ac.uk)

Maintaining biodiversity is crucial to the functioning of ecosystems and the delivery of ecosystem services[1], yet biodiversity is disappearing[2]. Mammals and birds, in particular, are diverse—comprising more than 15,000 living species—and are important ecological components in nutrient distribution, propagule (e.g., seed) dispersal, and as interactive connectors between species and habitats[3,4]. However, mammals and birds are subject to strong human pressure, leading to high extinction rates[5]. The diversity and extinction of mammals and birds has, to date, predominantly been studied according to taxonomy[6–8] and phylogenies[6,9,10]. However, species are also characterized by their traits—morphological, physiological, phenological or behavioral features measurable at the individual level[11], which can provide a more direct link than taxonomy or phylogeny to ecosystem processes and functions[4,12,13]. Traits jointly determine a species' ecological role[11,14] and thus combinations of traits are increasingly being used to summarize species' ecological strategies[15].

Mammals and birds exhibit strong ecological variation—from large hypercarnivores, to long-lived arboreal frugivores, to wide-ranging scavengers. Even so, many species share fundamentally similar strategies, such as flying insectivores (bats and birds), and traits often co-vary across species[16]. Mammal and bird species often compete for resources and thus face a broadly similar range of selection pressures (e.g., climatic events, predation, habitat change). Although similar selection pressures should lead to the adoption of comparable strategies (i.e., convergent evolution), evolutionary history[17] has applied constraints that will likely lead to divergence between mammals and birds. The contrast between the high ecological diversity but convergent strategies across mammal and bird species raises a fundamental question: how are ecological strategies presently organized across these two groups? We predict that mammals will show greater ecological diversity, given the rapid morphological, ecological, and phylogenetic diversification in terrestrial mammals during the Cenozoic that led to an expansion in mass by four orders of magnitude[18,19].

In addition, past and present human impacts have led to the accumulation of extinction debts—numerous species are already committed to extinctions that are yet to occur[20,21]. Extinction is a selective process because both extrinsic and intrinsic factors result in the non-random loss of species[22]. Thus, although exposure to threatening processes (extrinsic) is the ultimate cause of extinction, a species' ecological strategy (intrinsic) determines how well it is able to withstand the threats to which it is exposed[23]. Ecological strategies, and the individual traits that comprise them, can therefore be seen as adaptations to extrinsic rates of mortality[16,24]. For example, traits that confer ecological flexibility (e.g., generalist species) and allow populations to recover rapidly from depletion may offer a degree of protection from external threats[23], while large-bodied species generally have higher extinction risk than small-bodied species[3]. Employing probabilistic extinction frameworks allows us to evaluate the impact of paying off these extinction debts and forewarn us of potential ecological consequences, enabling us to act—before it is too late. For instance, when species become extinct locally and globally, their ecological strategies are lost[3,25], with potentially strong implications for ecosystem functions[2,15,25–27].

Here we focus on three primary research questions: (i) what are the major gradients across the diversity of mammal and bird ecological strategies? (ii) how do mammals and birds share ecological strategy space? (iii) how do projected extinctions affect ecological diversity when compared with random species loss? To explore species' ecological strategies, we ordinated (principal components analysis; PCA) all 15,484 living land mammals and birds based on five traits: body mass, litter/clutch size, habitat breadth, diet, and generation length[28]. The ordination of species

across this surface represents a 2-dimensional continuum, integrating ecological strategies within each of the five trait dimensions to form an ecological strategy surface, through which gradients can be identified[29,30]. We then constructed 5-dimensional ecological strategy spaces, via hypervolume estimation[31,32], for mammals and birds combined and separately. These ecological strategy spaces were compared to four alternative null models of multivariate trait variation, previously applied to plants[29], to understand strategy convergence across and between mammals and birds. Finally, we modeled the impact of future projected extinctions (i.e., evaluating the cost of the current extinction debt) on the global ecological strategy space. We forecasted the volume of ecological strategy space 100 years into the future, given extinction probabilities assigned to the IUCN Red List categories[33]. To put the loss of species with high extinction risk in perspective we compared the projected scenario to a randomized scenario, controlling for species richness. Overall, we summarize the ecological consequences of biodiversity loss.

Here, we find that the ecological diversity of mammals and birds is structured by life-history speed (fast–slow) and body mass (small–large) in one dimension, and diet (invertivore–herbivore) and habitat breadth (generalist–specialist) in the other dimension. We also show that the ecological strategy space currently occupied by mammals and birds is strongly restricted compared to null expectations. Moreover, we demonstrate that future projected extinctions result in a larger reduction of ecological strategy space than expected at random. Consequently, we find that paying off current extinction debts leads to a shift in the global composition of mammals and birds to smaller, faster-lived, more fecund, more generalist and preferentially insect-eating species, fundamentally restructuring life on our planet.

## Results

**Ecological strategy surface.** Despite high diversity in form and function of mammals and birds across the world, there are distinct patterns among trait combinations that define species' ecological strategies (Fig. 1). The first two principal components (PC1 and PC2) explained more than half (60%) of the total trait variation (Fig. 1), but there was some variation in all five principal components (Supplementary Fig. 1).

The primary axis of differentiation, PC1, integrates both a body mass gradient (body mass loading = 0.63) and the fast-slow continuum[16,23,34]—here, the trade-off between litter/clutch size (loading = −0.35) and generation length (loading = 0.58) (Fig. 1). Species with low PC1 values are therefore generally characterized by small body mass and fast life-history (short generation length, high litter/clutch size), e.g., shrews, rodents, passerines; whereas species with high PC1 values are distinguished by large body mass and slow life-history, e.g., elephants, rhinos, deer, pelicans (Fig. 1). PC1 therefore also reflects how quickly populations can recover from low levels, as slow life histories reduce the ability of populations to compensate for increased mortality[35]. Moreover, body mass relates to the contributions of species to multiple ecological functions, such as pollination[27,36], predation[37], herbivory[38], food-web structure[39] and seed-dispersal[27,40]. PC2 characterizes a gradient between invertivorous, habitat generalists (diet loading = −0.70, habitat breadth loading = −0.47) at low PC2 values, e.g., echolocating bats, swifts, seabirds; to herbivorous, habitat specialists at high PC2 values, e.g., marmots, duikers, rodents (Fig. 1). PC2 therefore reflects the trophic interactions of species with other food web components and, consequently, their effect on nutrient cycling[3,4]. PC2 also characterizes species responses to changes in resource availability and their capacity to adapt to environmental change, especially habitat

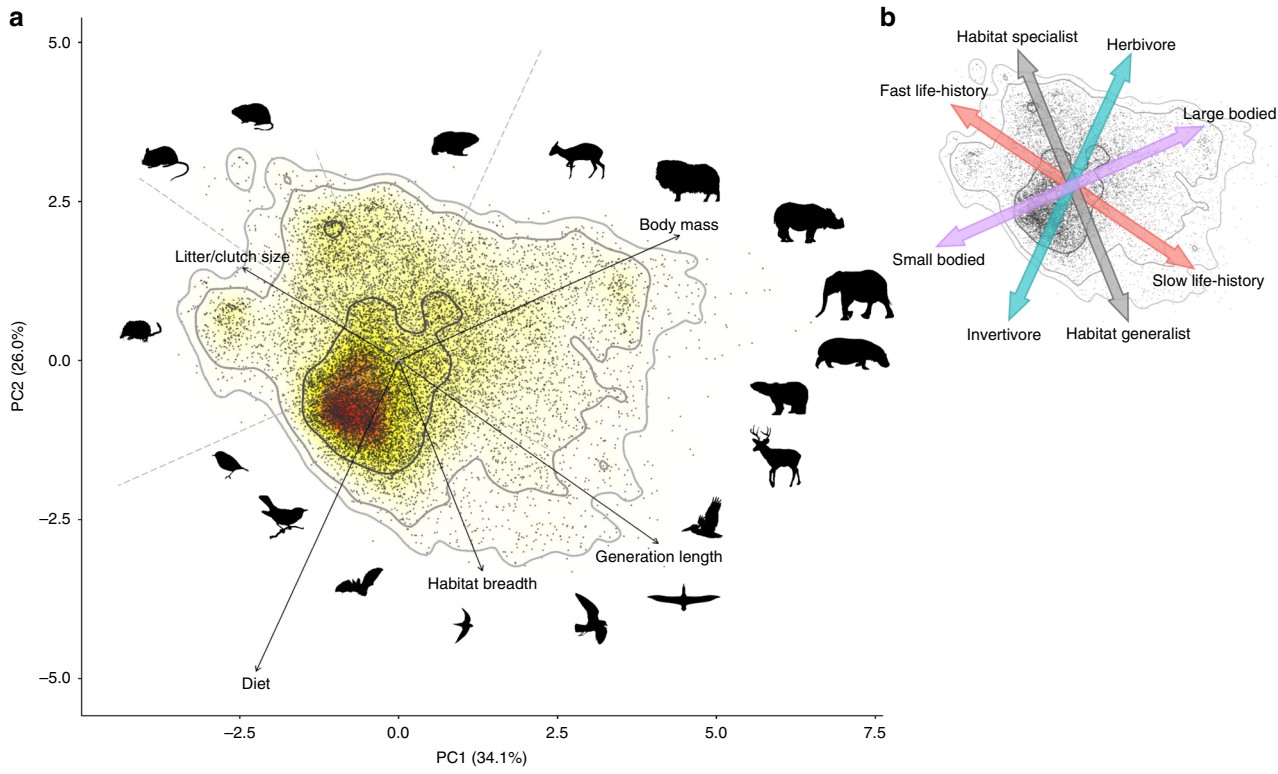

**Fig. 1** The ecological strategy surface for mammals and birds. **a** Projection of 15,484 living land mammal and bird species (dots) on the surface defined by principal component axes (PC) 1 and 2 (mean values across 25 imputed datasets; Supplementary Methods; Supplementary Fig. 2). Solid arrows indicate direction and weighting of vectors representing the five continuous traits analyzed (Supplementary Table 1 for loadings). Silhouettes show a selection of species characterizing the edges of strategy space (eight silhouettes were freely downloaded from PhyloPic www.phylopic.org, under CC0 1.0 Public Domain Dedication, while the rest were created in Inkscape by the authors; Supplementary Fig. 3 for species locations, scientific names and image sources). The color gradient indicates regions of highest (red) to lowest (white) occurrence probability of species across the ecological strategy surface, with contour lines indicating 0.5, 0.95, and 0.99 quantiles. Percentage values represent proportion of the total variation explained by each PC. To quantify diet, we extracted the dominant diet gradient across ten diet categories for all species, using a principal coordinates analysis (PCoA; Supplementary Fig. 4). **b** The ecological strategy surface is also illustrated with simplified gradients. Source data are provided as a Source Data file

modifications[41]. For instance, a broad habitat breadth confers greater ecological flexibility and thus the opportunity to shift resource use or distribution in response to environmental change. PC2 also generally distinguishes volant species from non-volant species (Supplementary Fig. 5c), not directly through their aerial mode (which was not used as a trait within our PCA), but via ecomorphological differences (reflecting previous results for mammals only[42]). The strongest correlations across the traits were between body mass and diet (Pearson's $r = -0.45$), body mass and generation length ($r = 0.41$), and generation length and litter/clutch size ($r = -0.34$) (Supplementary Fig. 6). The weakest correlations were between diet and litter/clutch size ($r = -0.02$), diet and generation length ($r = 0.06$), and body mass and habitat breadth ($r = 0.08$).

**Ecological strategy space**. We further find that the ecological strategy space currently occupied by mammals and birds is strongly restricted (9–62% occupation of null strategy spaces, all permutation tests $P \leq 0.001$; Supplementary Table 2) when compared to four alternative null models:[29] 1-traits uniformly distributed and independent from each other, approximately a hypercube (9% occupation); 2-traits normally distributed and independent from each other, approximately a hypersphere (37%); 3-traits distributed as observed and independent from each other (62%); 4-traits normally distributed and correlated as observed, approximately a hyperellipsoid (51%). Specifically, of all

possible trait combinations—null model 1 assumes any combination of trait values can arise and escape natural selection with equal probability[29,43]—only 9% are realized in contemporary mammal and bird ecological strategies and are therefore currently evolutionarily viable on Earth.

Our comparative analysis of mammals and birds reveals that the avian strategy space is more than a third smaller than that for mammals, despite birds being represented by around double the number of species (10,252 birds occupy a volume of 534 $SD^5$, while 5232 mammals occupy 881 $SD^5$ in volume) (Fig. 2). This contrast means that birds (19.2 species $SD^{-5}$) are more than three times more concentrated within their ecological strategy space than mammals (5.9 species $SD^{-5}$), indicating high strategy convergence and suggesting that ecological diversity and taxonomic diversity could be generated by different processes. We also observe low overlap between mammals and birds in strategy space (Supplementary Fig. 5a), with mammals and birds overlapping across 31% (intersection volume = 332 $SD^5$) of the total combined strategy space (combined volume = 1084 $SD^5$). Birds occupy 19% of space unoccupied by mammals (unique volume = 202 $SD^5$) and mammals 51% of the space unoccupied by birds (unique volume = 549 $SD^5$) (Fig. 2). Mammals therefore show a greater range of ecological modes, which we hypothesize indicates both greater net evolutionary change—the dissimilarity between species regardless of the evolutionary pathways—and possibly greater ecological adaptive potential, which should enhance the probability that at least some species will survive

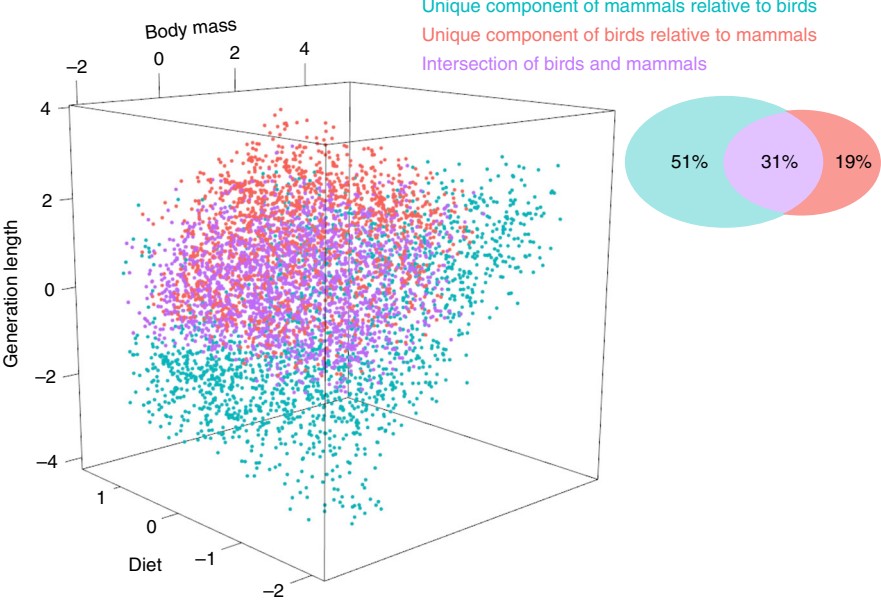

**Fig. 2** Overlap between mammal and bird ecological strategy spaces. The separation (unique components) and overlap (intersection) of 15,484 living land mammal and bird species across ecological strategy spaces (hypervolumes). The two ecological strategy spaces (one for mammals, one for birds) are constructed on the basis of the five z-transformed traits, although only the three traits with the highest loadings across the ecological strategy surface are used for visualization: $\log_{10}$(body mass), $\log_{10}$(generation length) and diet (Fig. 1a; Supplementary Table 1). The Venn diagram shows the percentage of the total combined volume occupied by each component (the percentages sum to 100 before rounding). Source data are provided as a Source Data file

into the future[44]. Although, the adaptive potential of mammals and birds will depend on the specific nature and types of selection pressures. Thus, in an adaptation context, we suggest that mammals show a greater range of specialization and adaptation, enabling them to persist and compete in dynamic environments, whereas birds have converged on a more generalized strategy (i.e., a diurnal, volant, invertivorous strategy; Fig. 1). The high convergence and generalized strategy of birds could be facilitated by their ability to fly (reflected by the high convergence shown by bats; Supplementary Table 3), allowing volant species to escape from disturbances[45] and competition.

**Projected extinctions**. We contrast projected and randomized extinction scenarios. For the projected extinction scenario, we assigned extinction probabilities to IUCN Red List categories, for example 66.7% of Endangered species and 10% of Vulnerable species went extinct for each simulation[33]. The randomized extinction scenario selected an equivalent number of species for extinction over the next 100 years, but randomly with respect to species identity and traits. We replicated the projected and randomized scenarios 999 times each. We find that the global ecological strategy space contracts more than expected at random under the projected extinction scenario (Kolmogorov-Smirnov test: randomized extinction mean = 1058 $SD^5$, projected extinction mean = 1021 $SD^5$; $D = 0.77$, $P < 0.001$) (Fig. 3). We also forecast over double the loss of ecological diversity over the next 100 years than expected at random (randomized compared to observed, effect size = −25.2 [95% CI: + 6.0, −57.7] $SD^5$; projected compared to observed, effect size = −62.5 [−34.3, −91.5] $SD^5$; Fig. 3). Thus, the ecological, and potentially functional, consequences of the projected extinctions are greater than would be expected under random species loss.

After 100 years of projected extinctions, the global composition of mammals and birds is predicted to shift to smaller (permutation test: body mass observed mean = 70.3 g, body mass projected mean across runs [minimum–maximum across

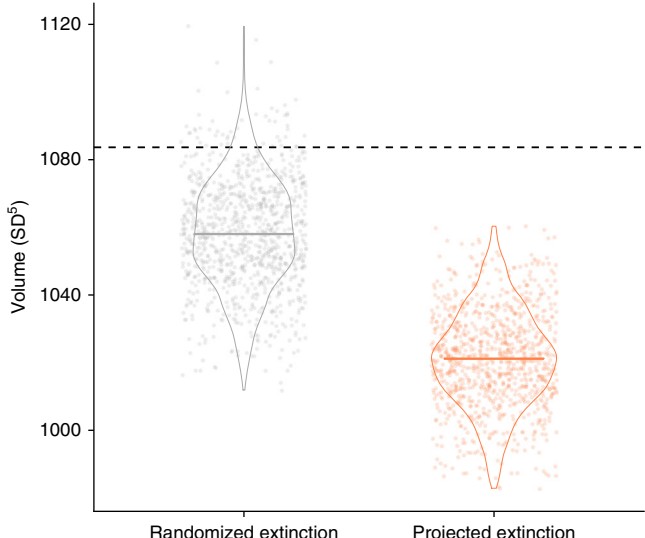

**Fig. 3** The ecological strategy space for mammals and birds under 100-year extinction scenarios. The dashed horizontal line indicates the observed ecological strategy space (hypervolume). For both scenarios we include jittered points for each of the 999 replicates, violins of data density and a central thick line of the mean. Due to the stochastic nature of the hypervolume algorithm[32], the ecological strategy space may increase as species are removed. 1095 mammal and bird species are lost under both the projected and randomized extinction scenarios, reflecting the probabilistic extinctions based on the IUCN threat categories (Methods). Source data are provided as a Source Data file

replicates] = 64.1 g [63.4–64.7 g]; $P \leq 0.001$), faster-lived (generation length observed mean = 4.27 years, projected mean = 4.22 years [4.21–4.23 years]; $P \leq 0.001$), more fecund (litter/clutch size observed mean = 2.51, projected mean = 2.55 [2.54–2.56]; $P \leq$

0.001), more generalist (habitat breadth observed mean = 3.23, projected mean = 3.32 [3.31–3.33]; $P \leq 0.001$) and more invertivorous species (diet observed mean = −0.00032, projected mean = 0.0012 [0.00087–0.0014]; $P \leq 0.001$) (Supplementary Figs. 5f and 7). These shifts are relatively large for the species pool and temporal scale investigated, for example, Davis et al.[10] showed that current median mammal body mass is 14% lower than during the Last Interglacial (~130,000 years ago), while we predict an extra 25.2% (23.9–25.8%) reduction in median mammal body mass over the next 100 years from the current level. These declines in body mass equate to a reduction rate of −0.00011% per year between the Last Interglacial and now, compared to a predicted reduction rate of −0.25% (−0.24 to −0.26%) per year between now and the next 100 years.

## Discussion

Despite high diversity across mammals and birds, we find a limited set of strategies that allow mammals and birds to survive natural selection, physiological challenges, and competitive exclusion currently on Earth. In particular, birds occupy a third less strategy space than mammals, despite around double the number of species. This supports our suggestion that rapid mammalian diversification during the Cenozoic led to high mammal ecological diversity[18,19], but limited taxonomic diversity. More generally, some trait combinations may be unobserved because they are non-viable (physically impossible, e.g., large-bodied and short generation length, or maladaptive), whereas others may be viable but not present within living species[30]. Strategies that are viable but are not currently realized could be due to a number of reasons, including: evolutionary factors (e.g., never evolved), ecological factors (e.g., competitively inferior strategies, strategies incapable of persisting within the current environment[30], such as extinct species), or they could occur in taxa not included in our analyses. Hence, as more trait data becomes available, comparative analyses among more distantly related taxa (e.g., all tetrapods, all vertebrates) will become possible, ultimately leading to a wider understanding of ecological strategy differentiation across species[46].

We forecast a substantial ecological downsizing for mammals and birds, supporting recent findings[3,5,26]. Ecological downsizing can entail the loss of unique ecological functions[2,25,47] and can impact ecosystem structure, function, and biogeochemical cycles[48,49]. Hence, downsizing could be a driver, as well as a consequence, of global change with implications for the long-term sustainability of ecological and evolutionary processes[2,10,19,48]. Here, we reveal that this extinction-driven shift in body mass extends to additional traits: generation length, litter/clutch size, habitat breadth, and diet, with further potential ecological consequences. For example, the predicted shortening of generation length could impact the timing and stability of ecological processes, such as scavenging. Among living vertebrates, only vultures are obligate scavengers[50]. Vultures are slow-lived (long generation length, low clutch size), highly threatened and are fundamentally involved in the scavenging of carrion in large packages[50]. Thus, the predicted loss of many vulture species (e.g., 8 are Critically Endangered) could have significant implications for scavenging and the spread of disease, as the initial loss of the most important species can cause rapid declines in ecosystem processes[51]. In addition, the predicted shift towards insect-eating species could potentially increase the susceptibility of the global species pool to specific threats, such as land use intensification or insect declines. For instance, insectivorous birds are less resilient to high-intensity than low-intensity land use[52], thus future land intensification could lead to further extinctions. Overall,

species' ecological strategies are intrinsically linked to extinction, and extinction to species' ecological strategies.

We demonstrate that the projected loss of mammals and birds will not be ecologically random, but a selective process across strategy space, where specific ecological strategies (e.g., slow-lived scavengers, herbivores, habitat specialists) will be filtered out; although, these directional changes could be directly or indirectly related to body mass, as many traits co-vary across species[16]. For example, diet and generation length were moderately correlated with body mass. Selection on body mass could therefore act as an extinction filter[3,5], driving shifts in the associated traits. Yet, body mass-associated extinction is likely to have further ecological consequences, as outlined above, due to the combinatory nature of traits (selection occurs on species' ecological strategies). In addition, we predict strong shifts in traits that are generally unrelated to body mass, such as habitat breadth and litter/clutch size. We therefore suggest that the ecological implications of the extinction debt go beyond body mass and emphasize that additional traits could have important roles in the process of extinction and selection.

There could also be additional impacts on species' ecological strategies not captured by our analyses. For instance, although we have summarized the breadth of a species' habitat use, which should confer a species' capacity to adapt to environmental change[41], habitat identity could also play an important role in a species' ecological strategy and function. We therefore suggest that further studies are needed to evaluate the fine-scale and spatial changes associated with paying off the extinction debt, as well as to establish the mechanisms leading to the compositional shifts in the ecological strategies of species quantified here.

The future defaunation explored here also shows parallels to historic extinction events, such as the late Quaternary extinctions, which likely disrupted species interactions, reduced long-distance seed dispersal, and fundamentally restructured energy flow and nutrient cycling through communities[26,53–55]. Moreover, a growing number of studies support the hypothesis that the late Quaternary extinctions had cascading effects on small vertebrates and plant community biodiversity and function, resulting in ecosystem shifts comparable in magnitude to those generated by climatic fluctuations[48,49,56]. Thus, the implications of the projected ecological impacts outlined here are extensive and complex.

While millennial-scale human pressures could have already filtered out the vast majority of sensitive species[5,57,58], we show that recent human activities might have generated an extinction debt with the capacity to non-randomly restructure mammals and birds on Earth, with potentially severe ecological consequences. Extinction debts were previously viewed as tragic, deterministic inevitabilities[20], but they can also be seen as opportunities for targeted conservation actions. As long as a species that is projected to become extinct persists, there is time for conservation action, such as habitat restoration or population management. For example, in the Amazon, recolonization due to forest regrowth slowed extinction rates and reduced the extinction debt for birds in rain forest fragments[59].

Here, we highlight that continuing to protect the most at risk species could help to preserve a diversity of ecological strategies, which could be important for ecosystems coping with environmental change[51], and maintaining ecosystem functionality. Moreover, we suggest that greater consideration of the ecological importance and diversity of mammals and birds could benefit conservation planning. Our work therefore underlines the multidimensionality of biodiversity and suggests that analyses of conservation prioritization across dimensions could be increasingly important into the future[15,60,61]. Finally, forecasting the loss of ecological diversity and the associated functional consequences should improve our ability to predict

and mitigate future responses that sustain ecosystems in the long-term.

## Methods

**Summary**. In brief, using five traits, we built an ecological strategy surface (2-D), via a PCA, and ecological strategy spaces (5-D), via hypervolume estimation. All analyses were carried out using R version 3.5.1 (ref.[62]).

**Traits**. We used five traits: body mass, litter/clutch size, habitat breadth (number of IUCN habitats listed as suitable), generation length and diet (the dominant diet gradient across ten diet categories for all species, see below; Supplementary Fig. 4) for 5232 mammal and 10,252 bird species. These traits reflect the resource acquisition, utilization and release by species and thus summarize a species' ecological strategy[28,63,64]. We extracted trait data for body mass, litter/clutch size and habitat breadth from our recently compiled—from four main sources[14,65–67]—database for mammals and birds[28]. For full details on the compilation of these three traits see Cooke et al.[28]. Generation length for birds was supplied by BirdLife. For mammals we obtained generation length values for mammals from a published dataset[66], although we corrected three mammal generation length observations that have since been found to be anomalous:[68] *Cephalophus adersi*, *Cephalophus leucogaster*, and *Cephalophus spadix*.

We removed four species from the trait dataset that have been confirmed as globally extinct since the trait data were compiled in 2016: Guam Reed-warbler *Acrocephalus luscinius* (last seen 1969), Bramble Cay melomys *Melomys rubicola* (last seen 2009), Christmas Island pipistrelle *Pipistrellus murrayi* (last seen 2009) and Bridled White-eye *Zosterops conspicillatus* (last seen 1983).

For diet, we calculated a continuous measure of a species' diet. Raw diet information was available as semi-quantitative records (percentage use of ten different dietary categories)[14]. To convert this information into a continuous measure, we first calculated Gower distances between species based on the diet data, gowdis() function in the FD package[69]. We then performed a principal coordinates analysis (PCoA) on the Gower distances, dudi.pco() function (ade4 package[70]). PCoA rotates the matrix of Gower distances to summarize inter-species (dis)similarity in a low-dimensional, Euclidean space[71]. Thus, PCoA does not change the positions of the species relative to each other but changes the coordinate system. Trait space and hypervolume analyses assume that all axes contribute equally to distances and volumes[31]. Thus, only the first principal component from the diet PCoA was used in the trait space and hypervolume analyses, so that each trait dimension had equal weight (although see the Supplementary Methods and Supplementary Fig. 11, where both the first and second principal components were used). The values yielded by the first principal component of the PCoA serve as synthetic trait values (i.e., new trait values based on the relative importance of diet categories in the initial dataset) and are referred to as 'diet'. Diet explained 36.2% of the variation across the diet categories and was predominantly loaded positively on invertebrates (PCoA loading = 3.69) and negatively on plant material (−1.66), fruit (−1.18), and seed (−0.80) (Supplementary Fig. 4); thus representing a gradient from invertivore to herbivore, reflecting previous diet ordination for mammals only[72].

Trait data were transformed where it improved normality: $\log_{10}$ for body mass, generation length and litter/clutch size; square root for habitat breadth; and all traits were standardized to zero mean and unit variance (z-transformation). Transformation and standardization to unitless coordinates is recommended for trait analyses[46,73] and hypervolume calculations[74].

**Trait imputation**. Trait data were not available for all species. Overall 12% of trait values were missing. The common practice of using only species with complete data (data-deletion approach) not only reduces sample size and consequently the statistical power of any analysis, but may also introduce bias[75,76]. Moreover, missing data would restrict the dimensionality of our analysis, as any species with at least one missing trait value cannot be used for hypervolume estimation, because an n-dimensional object is not well defined in fewer than n dimensions[74]. Instead, to achieve complete species-trait coverage we imputed missing data for litter/clutch size (42% imputed), habitat breadth (10%), diet (8%), and generation length (0.2%). Body mass data had complete species coverage. We used Multivariate Imputation with Chained Equations (MICE), based on the ecological (the transformed traits) and phylogenetic (the first ten phylogenetic eigenvectors extracted from trees for birds[77] and mammals[78]) relationships between species[28]. MICE has been shown to have greater accuracy, improved sample size and smaller error and bias than single imputation methods and the data deletion approach[75,76]. The data deletion approach was performed for comparative purposes (8294 species; Supplementary Fig. 8). To generate imputed values, we used the mice() function from the mice package[79].

To capture the uncertainty in the imputation process we imputed 25 trait datasets (Supplementary Fig. 2). These imputed datasets are based on the same input trait data, but differ in their estimations for the missing-data. Where possible we performed our analyses across the 25 imputed datasets (Fig. 1). However, utilizing the multiple datasets was not possible for the hypervolume analyses, due to the computational cost of the analyses (each hypervolume analysis took upto a day to run on a computer with an Intel Xeon CPU E5-2407 0 @ 2.2 GHz processor

and 96GB of RAM, thus running multiple analyses 25 times each was unfeasible). Instead, for the hypervolume analyses, we used a single, randomly selected, imputation dataset.

**Ecological strategy surface**. We built an ecological strategy surface (2-D) from the transformed and standardized traits via a PCA, using the princomp() function in the vegan package[80] (Fig. 1). The ordination of species across this surface represents a two-dimensional continuum, integrating ecological strategies within each of the five trait dimensions (i.e., creating an ecological strategy surface).

We used multivariate kernel density estimation to calculate the occurrence probability of given combinations of trait values (probability contours) across the ecological strategy surface[29], via the kde() function (ks package[81]). We extracted contours at the 0.5, 0.95, and 0.99 quantiles of the probability distribution (Fig. 1). Because results depend on the choice of the bandwidth used for the smoothing kernel, we used unconstrained bandwidth selectors[82]. Specifically, we used the sum of asymptotic mean squared error pilot bandwidth selector[83], through the Hpi() function in the ks package[81].

**Ecological strategy space**. To evaluate the ecological strategy spaces of mammals and bird combined, and separately, we constructed trait hypervolumes. One of the major advantages of the hypervolume approach is that it can accurately measure the volume of a high-dimensional shape that may include holes, disjunctions or other complex geometrical features[31,74], and thus hypervolumes model multi-dimensional spaces better than linear and continuous dimensions, such as convex hulls[84]. Moreover, hypervolumes are not as sensitive to outliers as convex hulls[74,84] and do not assume any parametric probability distribution[31,74]. To build our hypervolumes we used the one-class support vector machine (SVM) estimation method[31]. SVM provides a smooth fit around data that is insensitive to outliers, yields a binary boundary classification ('in' or 'out'), is invariant to rotational transformation (i.e., correlations between axes), and is computationally viable in large datasets and high-dimensional hyperspaces[31]. SVM is the most appropriate hypervolume method when extreme values in the observed data are thought to represent the true boundaries of the data[31], as is the case here. However, the principal disadvantage is that the boundaries of the hyperspace (and therefore volume) can change non-monotonically when species are removed (see Extinction scenarios), due to the stochastic nature of the SVM algorithm[32]. In other words, the volume can increase when species are removed, due to the stochastic re-drawing of the hyperspace boundaries. We calculated the observed hypervolume based on the transformed and standardized traits using the hypervolume_svm() function in the hypervolume package[32]. Conversion to unitless coordinates (here z-transformation) is required so that volumes and overlaps can be defined[31,74]. The units of the hypervolumes are reported as the standard deviations of centered and scaled transformed trait values, raised to the power of the number of dimensions ($SD^{number\ of\ dimensions}$).

The observed hypervolumes were compared to four alternative null models of multivariate variation of the transformed traits (see[29] for full null model specifications). To compare the hypervolumes, we calculated the occupation by the observed ecological strategy space of the mean of 999 strategy spaces generated from the assumptions of each null model (Monte-Carlo permutations), with the as.randtest() function (ade4 package[70]).

Null model 1: Species traits vary independently and each of them comes from a uniform distribution[29]. This null model assumes that each of the traits represents an independent axis of specialization and that the occurrence of extreme and central values is equally probable[29].

Null model 2: Species traits vary independently and each of them comes from a normal distribution[29]. This null model assumes that all traits evolve independently, as in null model 1, but extreme trait values are selected against during evolution[29].

Null model 3: Species traits vary independently but—unlike in the previous null models—there is no assumption about the distribution of trait variation; each trait varies according to the observed univariate distributions[29].

Null model 4: Species traits are normally distributed and follow the estimated correlation structure of the observed dataset[29]. This null model assumes that there are less than six independent axes of specialization and that extreme values are selected against[29].

**Extinction scenarios**. To test the impact of future projected extinctions over the next 100 years, we assigned extinction probabilities to the IUCN Red List categories:[33] 0.999 for Critically Endangered (CR), 0.667 for Endangered (EN), 0.1 for Vulnerable (VU), 0.01 for Near Threatened (NT) and 0.0001 for Least Concern (LC) species. In addition, 13% of mammals (665 species) and 1% of birds (59 species) are categorized as Data Deficient (DD). DD species were, for simplicity, treated as LC (i.e., assigned them an extinction probability of 0.0001)[34,85]. For our dataset this results in the loss of 380 CR species (99.9%), 576 EN species (66.7%), 125 VU (10%), 13 NT (1%), and 1 LC/DD species (0.01%) (total = 1095 species). Although we also provide alternative analyses where we (i) removed DD species and (ii) assigned DD species an average predicted extinction probability of 0.277 (Supplementary Methods). We also show the distribution of the IUCN Red List categories across the ecological strategy surface (Supplementary Fig. 5f).

We compared these projected extinctions to a null model based on randomized species extinctions, where an equivalent number of species go extinct over the next 100 years (1095 species) but randomly with respect to species identity and traits. We replicated both the projected and randomized scenarios 999 times. To evaluate the difference between the projected and randomized extinction scenarios we used a Kolmorgorov–Smirnov test with the ks.test() function (stats package[62]). We also calculated absolute effect sizes as observed volume—randomized volume and observed volume—projected volume, with 95% confidence intervals of the differences. To assess shifts in the trait distributions we used permutation tests, via the as.randtest() function (ade4 package[70]) (Supplementary Fig. 7).

**Sensitivity.** Overall our results and conclusions were qualitatively similar (i) with and without imputed trait data (Supplementary Figs. 2, 8, 9 and 10), (ii) when including the first or the first and second principal components from the diet PCoA (Supplementary Figs. 4 and 11), (iii) with and without DD species (Supplementary Figs. 12 and 13), and (iv) when assigning DD species an extinction probability of 0.0001 or 0.277 (Supplementary Figs. 14 and 15). Further information on these analyses is provided in the Supplementary Methods.

**Reporting summary.** Further information on research design is available in the Nature Research Reporting Summary linked to this article.

## Data availability

The trait data were principally extracted from Cooke et al.[28], which was compiled from four main databases[14,65–67] and is available on figshare (https://figshare.com/articles/Global_tradeoffs_of_functional_redundancy_and_functional_dispersion_for_birds_and_mammals/5616424; file: trait_data.csv). Generation length for mammals[66] (https://datadryad.org/resource/doi:10.5061/dryad.gd0m3) and raw diet data[14] (https://figshare.com/articles/Data_Paper_Data_Paper/3559887) were additionally compiled here. Generation length for birds was supplied by BirdLife but restrictions apply to these data, which were used under license for the current study. However, these data can be manually downloaded from the BirdLife website (http://datazone.birdlife.org/species/search). The code and data (without generation length due to data restrictions) to replicate our analyses is available on Github: https://github.com/03rcooke/hyper_pca. In addition, the source data underlying Figs. 1, 2 and 3 and Supplementary Figs. 1, 2, 3, 4, 5, 6, 7, 8, 9, 10, 11, 12, 13, 14, 15, 16 and 17 are provided as a Source Data file.

## Code availability

The simplified code, as an R notebook, is available on Github: https://github.com/03rcooke/hyper_pca.

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

## Acknowledgements

This work was supported by a scholarship (awarded to R.S.C.C.) co-funded by the SPITFIRE Doctoral Training Partnership (supported by the Natural Environmental Research Council, grant number: NE/l002531/1) and the University of Southampton. We thank A. Chapman, L. Graham, and R. Holland for comments, discussions, and suggestions that greatly improved the manuscript.

## Author contributions

R.S.C.C., F.E., and A.E.B. formulated the study, R.S.C.C. developed and implemented the analyses and wrote the first draft. All authors contributed to interpreting the results and the editing of manuscript drafts.

## Additional information

**Competing interests:** The authors declare no competing interests.

