## [Peer Review File · Nature Communications]

Reviewers' Comments:

Reviewer #1:

Remarks to the Author:

This is an extremely interesting and well-crafted paper, which aims at identifying current ecological strategies of mammals and birds and project them into the future in order to predict losses of these strategies. The analysis is solid and well explained, and the discussion mirrors the results, with good explanations and consideration of existing literature. I only have a few concerns on the methods.

First of all, the authors treated DD species as LC in order to be more conservative, but I am not convinced of this explanation. Several papers that attempted to predict the conservation status of DD species (eg. Bland et al. 2015 Cons Bio, Morais et al. 2013 Biol Cons) show that the majority of them could be actually classified as threatened. Most of these species have very restricted distribution and would probably fall into a threatened category.

Did the authors use raw data from the different databases for multiple imputations (e.g. EUTHERIA)? Or did they use a mix of raw and averaged data? Or only averaged data? The first method is the most appropriate one, and this will affect the final estimate of the value of the trait. In addition, I couldn't easily find the names of the 25 databases used that are mentioned in the SI.

Finally, some of the traits used in the analysis are usually correlated (e.g. high body mass and long generation length, long generation length and small litter/clutch size). Did the authors consider that in their analysis? I would at least discuss this in the main text.

Reviewer #2:

Remarks to the Author:

The manuscript "Projected losses of global mammal and bird ecological strategies" defines the multidimensional trait space occupied by mammals and birds, and describes the changes that this space is likely to experience given impending extinctions. The manuscript is timely and important – it parallels previous work on plants by Díaz and colleagues. It is presented in a concise and engaging way, it is well structured and well written. My comments are mostly medium to minor and nearly all related to the details of the methods.

Concerns:

Methods

- I am not convinced that using exclusively the first Principal Component from the diet PCoA is sufficient to capture overall diet variation, especially when combining taxa as diverse as birds and mammals. For instance, the functional importance and uniqueness of carnivory does not appear to be well captured. The authors mention that PC1 explains 36.2% of the variation across diet categories – what about the remaining variation? How important is it? What would happen to the overall multidimensional trait space if it included two diet axes?
- What is the effect of imputing 42% of litter/clutch size and, in addition, of doing so using a phylogenetic imputation method? Could this introduce a taxonomic and/or phylogenetic bias to the pattern of extinction-driven trait shift?
- Is habitat breadth an informative trait axis in this context? I understand the usefulness of a specialist

versus generalist axis but exclusively using the number of habitats occupied without any reference to the identity of those habitats may risk lumping species together with very different habitat requirements. How would the trait space look if the authors summarized habitat occupancy in a way akin to the way they summarized diet? If habitat categories have been taken from Wilman et al for mammals, proportional representation for those could be obtained using $1/(\text{number of habitats})$ across all habitats associated with a species.

- Considering all Data Deficient species as Least Concern may be a problematic assumption: it is often the case that species are Data Deficient because they are less widely-distributed and/or less common and, therefore, more rarely observed and documented. Moreover, if Data Deficient species lack data partly because of their traits (e.g. small body size), assigning them all a very small probability of extinction may lead to a biased pattern of extinction-driven trait change. Would the pattern of extinction-driven trait change be different if all Data Deficient species were labelled as Endangered or Critically Endangered?

Results

- What is the percentage of null mammal trait space occupied by birds (and vice versa)? In other words, are birds taking advantage of potentially unused mammal trait space (and vice versa)?
- Please clarify how trait volumes can increase given species extinctions

Discussion

- Are predicted shifts in traits other than body mass (e.g. generation length) due to their direct role as extinction filters or are those traits simply shifting because they covary with body mass?
- Please add a paragraph to mention and discuss the main assumptions and caveats in these analyses, e.g. with respect to the methodological issues raised above

NCOMMS-19-02120A

Projected losses of global mammal and bird ecological strategies

Reviewers' comments (black text)

Author responses (blue bold text, with manuscript text revisions in blue text)

Reviewer #1 (Remarks to the Author):

This is an extremely interesting and well-crafted paper, which aims at identifying current ecological strategies of mammals and birds and project them into the future in order to predict losses of these strategies. The analysis is solid and well explained, and the discussion mirrors the results, with good explanations and consideration of existing literature. I only have a few concerns on the methods.

Thank you very much for these very positive comments!

First of all, the authors treated DD species as LC in order to be more conservative, but I am not convinced of this explanation. Several papers that attempted to predict the conservation status of DD species (eg. Bland et al. 2015 Cons Bio, Morais et al. 2013 Biol Cons) show that the majority of them could be actually classified as threatened. Most of these species have very restricted distribution and would probably fall into a threatened category.

That is a good point! We agree that our treatment of DD species may not be conservative in terms of the results. While we previously viewed our approach as conservative based on the number of species extinction in the next 100 years (1,095 species), we have now assessed the impact of DD species on our findings. For this we ran two additional scenarios, one where we excluded DD species and one where we assigned an average predicted extinction probability to DD species (0.277), leading to an additional 201 species extinctions. For the predicted extinction probability we used the work of Bland et al. 2015, so thank you for bringing this to our attention. The two additional DD scenarios have now been described in the Supplementary (Supplementary pg. 8):

“Here, for simplicity, we treated DD species as LC (i.e., assigned them an extinction probability of 0.0001)³⁷⁻³⁹. However, to evaluate the impact of DD species we also implemented two alternative scenarios: 1) excluding DD species (Figures 12, 13); and 2) assigning an average predicted extinction probability to DD species (Figures 14, 15), based on previous work⁴⁰. To calculate an extinction probability for DD species we first calculated the average extinction probability of threatened species at the same ratio of CR:EN:VU (380:863:1254 species) as for the set of species for which threat categories are known⁴¹. Thus, the average extinction probability for threatened species was $0.433 = ((380 \text{ CR species} * 0.999) + (863 \text{ EN species} * 0.667) + (1254 \text{ VU species} * 0.1)) / (380 + 863 + 1254)$. We then did the same for non-threatened species $((1300 \text{ NT species} * 0.01) + (10963 \text{ LC species} * 0.0001)) / (1300 + 10963) = 0.001$. 64% of DD mammal species were previously predicted to

be threatened, using machine learning tools⁴⁰. As the majority of our DD species were mammals (665 DD mammal species, 59 DD bird species) we applied this value across our 724 DD species. So we multiplied the average extinction probability for threatened species (0.433) by the proportion of DD species predicted to be threatened (0.64)⁴⁰ and multiplied the average extinction probability for non-threatened species (0.001) by the proportion of DD species predicted to be non-threatened (0.36), and then summed the total extinction probability, resulting in an extinction probability of 0.277 for DD species. Thus, DD species were assigned an average predicted extinction probability that falls between VU and EN.”

We found strong agreement between the different treatments of DD species: Kolmogorov-Smirnov test D = 0.77 for DD species as LC, 0.81 excluding DD species, 0.80 using average extinction probability for DD species. Randomized compared to observed: effect size = -25.2 SD⁵ for DD species as LC, -27.2 SD⁵ excluding DD species, -28.1 SD⁵ using average extinction probability for DD species; projected compared to observed: effect size = -62.5 SD⁵ for DD species as LC, -67.3 SD⁵ excluding DD species, -68.9 SD⁵ using average extinction probability for DD species. All additional results reported in the Supplementary (Supplementary Figures 12, 13, 14 and 15).

Did the authors use raw data from the different databases for multiple imputations (e.g. EUTHERIA)? Or did they use a mix of raw and averaged data? Or only averaged data? The first method is the most appropriate one, and this will affect the final estimate of the value of the trait. In addition, I couldn't easily find the names of the 25 databases used that are mentioned in the SI.

The trait data used for this study were extracted from our previously published database (Cooke et al. 2019 GEB), which used primarily raw data (i.e., averaged data were generally avoided where possible), although for species that lacked body mass data the average body mass of congeneric or confamilial species was calculated (this was required so that all species overlapped in at least one trait dimension) (Cooke et al. 2019 GEB). For full details of the trait compilation process see the Supporting Information of Cooke, R. S. C., Bates, A. E. & Eigenbrod, F. Global trade-offs of functional redundancy and functional dispersion for birds and mammals. *Glob. Ecol. Biogeogr.* 28, 484–495 (2019).

We have edited the text to better explain where the trait data was obtained from (Supplementary pg. 3):

“We extracted trait data for body mass, litter/clutch size and habitat breadth from our recently compiled trait database for mammals and birds² (available from https://figshare.com/articles/Global_trade-offs_of_functional_redundancy_and_functional_dispersion_for_birds_and_mammals/5616424, file: trait_data.csv). For full details on the compilation of these three traits see Cooke et al².”

The use of averaged data, as you outline, could affect the imputation results, however the sensitivity tests show that the results are qualitatively similar when excluding species with imputed data (data-deletion approach; Supplementary Figure 8) and across the 25 imputed datasets (Supplementary Figure 2), which suggests the impact of averaged values is small.

The 25 imputed datasets generated do not have names, but are in fact repeated runs of the imputation procedure to account for the uncertainty in the imputation process. Apologies for this miscommunication. To better describe the imputed datasets we have edited the Supplementary to read (Supplementary pg. 5):

“To capture the uncertainty in the imputation process we imputed 25 trait datasets (Figure 2) (available on Github: https://github.com/03rcooke/hyper_pca/blob/master/data/df_tr_mi.rds). These imputed datasets are based on the same input trait data, but differ in the estimations for the missing-data. Where possible we performed our analyses across the 25 imputed datasets (main text Fig. 1). Utilizing the multiple datasets was not possible for the hypervolume analyses, due to the computational cost of the analyses (each hypervolume analysis took up to a day to run on a computer with an Intel Xeon CPU E5-2407 0 @ 2.2 GHz processor and 96GB of RAM, thus running multiple analyses 25 times each was unfeasible). Instead, for the hypervolume analyses, we used a single, randomly selected, imputation dataset.”

Hopefully this makes it clearer.

Finally, some of the traits used in the analysis are usually correlated (e.g. high body mass and long generation length, long generation length and small litter/clutch size). Did the authors consider that in their analysis? I would at least discuss this in the main text.

This is a really good point. We previously ran some preliminary correlation tests between the traits to better understand the relationships between each, but have now formally outputted the results as a correlation plot (Supplementary Figure 6) and have outlined the major correlations in the text (pg. 6):

“The strongest correlations across the traits were between body mass and diet (Pearson’s $r = -0.45$), body mass and generation length ($r = 0.41$), and generation length and litter/clutch size ($r = -0.34$) (Supplementary Figure 6). The weakest correlations were between diet and litter/clutch size ($r = -0.02$), diet and generation length ($r = 0.06$), and body mass and habitat breadth ($r = 0.08$).”

Plus, by using a PCA the correlated traits are decomposed into a set of uncorrelated principal components and thus highly correlated traits will load onto the same principal component. Although the use of highly correlated traits can overemphasize certain dimensions, the traits included all represent complementary aspects of a species’ ecological strategy (Cooke et al., 2019 GEB) and the correlations between the traits are generally low ($r < \pm 0.45$). In addition, the support vector machine algorithm used to

calculate the hypervolumes (i.e., ecological strategy space) is invariant to rotational transformation (i.e. correlations between axes). We have now described this in the Supplementary (Supplementary pg. 6):

“SVM provides a smooth fit around data that is insensitive to outliers, yields a binary boundary classification (‘in’ or ‘out’), is invariant to rotational transformation (i.e., correlations between axes), and is computationally viable in large datasets and high-dimensional hyperspaces¹⁵.”

Reviewer #2 (Remarks to the Author):

The manuscript “Projected losses of global mammal and bird ecological strategies” defines the multidimensional trait space occupied by mammals and birds, and describes the changes that this space is likely to experience given impending extinctions. The manuscript is timely and important – it parallels previous work on plants by Díaz and colleagues. It is presented in a concise and engaging way, it is well structured and well written. My comments are mostly medium to minor and nearly all related to the details of the methods.

Thank you for this positive feedback! Hopefully we have addressed your concerns and further assessed the methods.

Concerns:

Methods

- I am not convinced that using exclusively the first Principal Component from the diet PCoA is sufficient to capture overall diet variation, especially when combining taxa as diverse as birds and mammals. For instance, the functional importance and uniqueness of carnivory does not appear to be well captured. The authors mention that PC1 explains 36.2% of the variation across diet categories – what about the remaining variation? How important is it? What would happen to the overall multidimensional trait space if it included two diet axes?

Good point! We had the same concern, so we carried out a pilot analysis including two synthetic diet traits and found that the ecological strategy surface is qualitatively very similar including and excluding the second principal component from the diet PCoA (compare main text Fig. 1 and Supplementary Figure 11). We previously decided not to include the plot with diet PC2 in the Supplementary as it invalidates the equal weighting of the traits (by including two diet axes we essentially double-weight diet across the ecological strategy surface). However we agree that the representation of the surface with two diet axes is useful and informative, so we have now included it in the Supplementary (Supplementary Figure 11).

We also agree that carnivory is a functionally important, albeit low frequency, strategy. Although from visual inspection of the ecological strategy surface categorized by dietary guild (Supplementary Figure 5d) we decided that carnivores are well characterized on the ecological strategy surface (the vertebrate-eating dietary guild in green is clustered

separate from the other dietary guilds; Supplementary Figure 5d), and thus we believe the diet axis, combined with the other traits, captures the adaptations required for a carnivorous lifestyle.

We have now described this in the Supplementary (Supplementary pg. 4):

“Trait space and hypervolume analyses assume that all axes contribute equally to distances and volumes¹⁵. Thus, only the first principal component from the diet PCoA was used in the trait space and hypervolume analyses, so that each trait dimension had equal weight. Moreover, diet guild (see Categorical traits section) showed clear patterning across the ecological strategy surface, indicating that the use of a single diet axis sufficiently captured the variation in species diets (Figure 5d). For instance, carnivores show distinct separation on the ecological strategy surface (Figure 5d, see marginal plot on PC1), despite the low importance of carnivory in the PCoA (Figure 4). The values yielded by the first principal component of the PCoA serve as synthetic trait values (i.e., new trait values based on the relative importance of diet categories in the initial dataset) and are referred to as ‘diet’. Diet explained 36.2% of the variation across the diet categories and was predominantly loaded positively on invertebrates (PCoA loading = 3.69) and negatively on plant material (-1.66), fruit (-1.18) and seed (-0.80) (Figure 4); thus representing a gradient from invertivore to herbivore, reflecting previous diet ordination for mammals only¹⁶.

We also provide the ecological strategy surface when including two synthetic diet traits (first and second principal components from diet PCoA; Figure 4) for reference (Figure 11), which is very similar to when we only include one synthetic diet trait (main text Fig. 1).”

- What is the effect of imputing 42% of litter/clutch size and, in addition, of doing so using a phylogenetic imputation method? Could this introduce a taxonomic and/or phylogenetic bias to the pattern of extinction-driven trait shift?

We tested the effect of imputation on the ecological strategy surface implementing the data deletion approach (i.e., removing species with missing trait data; Supplementary Figure 8) and by comparing the surface across each of the 25 imputed datasets (Supplementary Figure 2). We show that the imputation does not qualitatively effect the results for the ecological strategy surface and have now added this to the Supplementary (Supplementary pg. 6):

“Overall our results and conclusions for the ecological strategy surface were similar (i) with and without imputed data (compare main text Fig. 1 and Figure 8; Figure 2), and (ii) with one or two synthetic diet traits (compare main text Fig. 1 and Figure 11).”

However we agree that the effect of imputation is important to evaluate, so we have now included supplementary results for the extinction analyses under the data deletion approach as well (Supplementary Figures 9 and 10). For the data deletion approach 514 species go extinct out of 8,294 total species leading to reduced effect sizes, however the main conclusions hold (Supplementary pg. 21):

“Kolmogorov-Smirnov test: observed extinction mean under the data deletion approach = 784 SD⁵, randomized extinction mean under the data deletion approach = 788 SD⁵, projected extinction mean under the data deletion approach = 759 SD⁵; D = 0.85, P < 0.001. Randomized compared to observed, effect size = +4.2 [95% CI: +23.9, -16.9] SD⁵, projected compared to observed, effect size = -24.5 [-6.4, -43.8] SD⁵.”

We have not performed the extinction analyses on each of the 25 imputed datasets to assess imputation uncertainty, as it is computationally intensive and we found strong agreement across the ecological strategy surface (Supplementary Figure 2). We have outlined this in the Supplementary (Supplementary pg. 5):

“Utilizing the multiple datasets was not possible for the hypervolume analyses, due to the computational cost of the analyses (each hypervolume analysis took up to a day to run on a computer with an Intel Xeon CPU E5-2407 0 @ 2.2 GHz processor and 96GB of RAM, thus running multiple analyses 25 times each was unfeasible). Instead, for the hypervolume analyses, we used a single, randomly selected, imputation dataset.”

- Is habitat breadth an informative trait axis in this context? I understand the usefulness of a specialist versus generalist axis but exclusively using the number of habitats occupied without any reference to the identity of those habitats may risk lumping species together with very different habitat requirements. How would the trait space look if the authors summarized habitat occupancy in a way akin to the way they summarized diet? If habitat categories have been taken from Wilman et al for mammals, proportional representation for those could be obtained using 1/(number of habitats) across all habitats associated with a species.

That’s an interesting point, unfortunately we do not currently have data on the proportional use of habitat types. Instead, habitat breadth was coded using the IUCN Habitats Classification Scheme (<http://www.iucnredlist.org/technical-documents/classification-schemes/habitats-classification-scheme-ver3>) and was quantified as the number of habitats listed as ‘Suitable’ for each species (Cooke et al., 2019 GEB). However we agree that habitat identity could be an important, and complementary, aspect of a species’ ecological strategy. As you outline our current measure is intended to summarise the breadth of a species habitat use from specialist to generalist. Our measure therefore differentiates ubiquitous generalists such as mule deer and wild boar, from the many specialists that are found in a single habitat type (~3,000 species).

We have now discussed the potential for including habitat identity in the main text (pg. 12):

“There could also be additional impacts on species’ ecological strategies not captured by our analyses. For instance, although we have summarized the breadth of a species habitat use, which should confer its capacity to adapt to environmental change⁴¹, habitat identity could

also play an important role in a species' ecological strategy and function. We therefore suggest that further studies are needed to evaluate the fine-scale and spatial changes associated with paying off the extinction debt, as well as to establish the mechanisms leading to the compositional shifts in the ecological strategies of species quantified here."

- Considering all Data Deficient species as Least Concern may be a problematic assumption: it is often the case that species are Data Deficient because they are less widely-distributed and/or less common and, therefore, more rarely observed and documented. Moreover, if Data Deficient species lack data partly because of their traits (e.g. small body size), assigning them all a very small probability of extinction may lead to a biased pattern of extinction-driven trait change. Would the pattern of extinction-driven trait change be different if all Data Deficient species were labelled as Endangered or Critically Endangered?

We agree that we had not fully evaluated the effect of DD species on our findings. To amend this we have undertaken two additional scenarios (also outlined in our response to Reviewer 1), one where we excluded DD species and one where we assigned an average predicted extinction probability to DD species (0.277). These are described in full in the Supplementary (Supplementary pg. 8):

"Here, for simplicity, we treated DD species as LC (i.e., assigned them an extinction probability of 0.0001)³⁷⁻³⁹. Although, to evaluate the impact of DD species we also implemented two alternative scenarios: excluding DD species (Figures 12, 13) and assigning an average predicted extinction probability to DD species (Figures 14, 15), based on previous work⁴⁰. To calculate an extinction probability for DD species we first calculated the average extinction probability of threatened species at the same ratio of CR:EN:VU (380:863:1254 species) as for the set of species for which threat categories are known⁴¹. Thus, the average extinction probability for threatened species was $0.433 = ((380 \text{ CR species} * 0.999) + (863 \text{ EN species} * 0.667) + (1254 \text{ VU species} * 0.1)) / (380 + 863 + 1254)$. We then did the same for non-threatened species $((1300 \text{ NT species} * 0.01) + (10963 \text{ LC species} * 0.0001)) / (1300 + 10963) = 0.001$. 64% of DD mammal species were previously predicted to be threatened, using machine learning tools⁴⁰. As the majority of our DD species were mammals (665 DD mammal species, 59 DD bird species) we applied this value across our 724 DD species. So we multiplied the average extinction probability for threatened species (0.433) by the proportion of DD species predicted to be threatened (0.64)⁴⁰ and multiplied the average extinction probability for non-threatened species (0.001) by the proportion of DD species predicted to be non-threatened (0.36), and then summed the total extinction probability, resulting in an extinction probability of 0.277 for DD species. Thus, DD species were assigned an average predicted extinction probability that falls between VU and EN."

We found strong agreement between the different treatments of DD species: Kolmogorov-Smirnov test D = 0.77 for DD species as LC, 0.81 excluding DD species, 0.80 using average extinction probability for DD species. Randomized compared to observed: effect size = -25.2 SD⁵ for DD species as LC, -27.2 SD⁵ excluding DD species, -28.1 SD⁵ using

average extinction probability for DD species; projected compared to observed: effect size = -62.5 SD^5 for DD species as LC, -67.3 SD^5 excluding DD species, -68.9 SD^5 using average extinction probability for DD species. All additional results reported in the Supplementary (Supplementary Figures 12, 13, 14 and 15).

Results

- What is the percentage of null mammal trait space occupied by birds (and vice versa)? In other words, are birds taking advantage of potentially unused mammal trait space (and vice versa)?

This is a really interesting question. However, it turns out the answer is quite complicated! Here is how we addressed this question:

First, we calculated the overlap between the observed bird strategy space and the null mammal strategy spaces. We find that birds are not completely nested within null mammal space. For example 2.6% of the observed bird ecological strategy space falls outside the null 1 mammal space (species traits vary independently and each of them comes from a uniform distribution) and 20.1% outside the null 2 mammal space (species traits vary independently and each of them comes from a normal distribution). This is due to birds having extreme values compared to mammals for some traits (i.e., trait values that fall outside the range of mammals). For instance, maximum litter/clutch size is higher for birds than mammals (Black-bellied Crimson Finch *Neochmia phaeton*), as is maximum generation length (Light-mantled Albatross *Phoebastria palpebrata*).

Although the above does not directly answer your question it is currently not possible to calculate the overlap between three hypervolumes (i.e., the overlap between observed mammal space, observed bird space and null mammal space). Thus, we instead compare the observed strategy spaces to the null strategy spaces for mammals and birds separately and combined (i.e., a shared null strategy space) (Supplementary Table 2). Fig. 2 in the main text shows that birds occupy 19% of ecological strategy space unoccupied by mammals, and if possible to calculate, we would expect this overlap to be reflected within the null mammal space (but slightly reduced due to the extension of birds outside of null mammal space). We therefore suggest that, taken together, Fig. 2 and our null model results (Supplementary Table 2) can help to elucidate the separation of mammals and birds in observed and null ecological strategy space. To highlight this interpretation we have added the following to the Supplementary information (Supplementary pg. 17):

“We also tested across taxonomic groups, i.e., birds within mammal null strategy space. We found that birds are not completely nested within the null strategy space for mammals (and vice versa), due to extreme trait values that fall outside of the range of trait values for the other taxa (e.g., maximum generation length and litter/clutch size is greatest for birds, whereas maximum body mass and habitat breadth is greatest for mammals). Thus the different taxonomic groups do not completely share the same potential suite of trait

combinations. In multi-dimensional space we find that 2.6% of the observed bird strategy space is unique compared to the null 1 mammal space, whereas 22.8% of the observed mammal space falls outside the null 1 bird space.”

- Please clarify how trait volumes can increase given species extinctions

The calculation of volume, using the hypervolume approach, is not dependent on the number or density of species in ecological strategy space, it is only concerned with measuring the borders of the space. The SVM algorithm stochastically measures the volume, which can, by chance, lead to a larger volume even after species are removed, especially if few species are lost at the edges of ecological strategy space. This however, makes the consistently lower volume under the projected scenario all the more striking.

We have now expanded the Supplementary to read (Supplementary pgs. 6/7):

“To build our hypervolumes we used the one-class support vector machine (SVM) estimation method¹⁵. SVM provides a smooth fit around data that is insensitive to outliers, yields a binary boundary classification (‘in’ or ‘out’), is invariant to rotational transformation (i.e., correlations between axes), and is computationally viable in large datasets and high-dimensional hyperspaces¹⁵. SVM is the most appropriate hypervolume method when extreme values in the observed data are thought to represent the true boundaries of the data¹⁵, as is the case here. However, the principal disadvantage is that the boundaries of the hyperspace (and therefore volume) can change non-monotonically when species are removed (see Extinction scenarios), due to the stochastic nature of the SVM algorithm³². In other words, the volume can increase when species are removed, due to the stochastic re-drawing of the hyperspace boundaries.”

Discussion

- Are predicted shifts in traits other than body mass (e.g. generation length) due to their direct role as extinction filters or are those traits simply shifting because they covary with body mass?

Thanks for raising this point. We agree that the mechanisms for the shift in traits could be directly or indirectly related to body mass and we have now extended the main text to discuss this (pgs. 11/12):

“We demonstrate that the projected loss of mammals and birds will not be ecologically random, but a selective process across strategy space, where specific ecological strategies (e.g., slow-lived scavengers, herbivores, habitat specialists) will be filtered out; although, these directional changes could be directly or indirectly related to body mass, as many traits co-vary across species¹⁶. For example, diet and generation length were moderately correlated with body mass. Selection on body mass could therefore act as an extinction filter^{3,5}, driving shifts in the associated traits. Still, body mass associated extinction is likely to have further ecological consequences, as outlined above, due to the combinatorial nature

of traits (selection occurs on species' ecological strategies). In addition, we predict strong shifts in traits that are generally unrelated to body mass, such as habitat breadth and litter/clutch size. We therefore suggest that the ecological implications of the extinction debt go beyond body mass and emphasize that additional traits could have important roles in the process of extinction and selection."

We have also provided a correlation plot (Supplementary Figure 6) to help understand each traits relationship to body mass

- Please add a paragraph to mention and discuss the main assumptions and caveats in these analyses, e.g. with respect to the methodological issues raised above

Instead of a single paragraph we have now added text describing the caveats of work throughout the main paper and Supplementary (as described per methodological issue raised above).

We have also added the following to the Methods to flag the major methodological considerations evaluated in our analysis and signpost readers to the further work carried out in the Supplementary (pgs. 13/14):

"Overall our results and conclusions were qualitatively similar (i) with and without imputed trait data (Supplementary Figures 2, 8, 9 and 10), (ii) when including the first or the first and second principal components from the diet PCoA (Supplementary Figures 4 and 11), (iii) with and without DD species (Supplementary Figures 12 and 13), and (iv) when assigning DD species an extinction probability of 0.0001 or 0.277 (Supplementary Figures 14 and 15)."

Reviewers' Comments:

Reviewer #2:

Remarks to the Author:

I have now reviewed the revised version of the manuscript "Projected losses of global mammal and bird ecological strategies", paying particular attention to the point-by-point response to the comments raised by both reviewers on the previous version. I commend the authors on a truly great effort revising this manuscript – I think it is an improvement on an already excellent manuscript and it will provide an extremely useful contribution to the field.

I only have very minor edits and a couple of suggestions (see below) – I will leave it to the editor and authors to decide whether these might be useful or not.

Finally, I was wondering if the authors were planning on releasing data on their estimated ecological trait spaces (e.g. species-level values for Fig. 1's PC1 and PC2)? I am unsure whether this is a requirement or not in Nature Communications; however, I thought I would mention that I believe it would provide a fantastic resource for the research community and would undoubtedly stimulate many follow-up studies.

Best,
Giovanni Rapacciuolo

Line edits (lines)

Abstract

27 – Suggest adding "future" after "projected"

Introduction

51 – correct to "a species' ecological role"

62-68 – I suggest moving the sentence with the prediction ("We predict that mammals will show...") after that with the question ("how are ecological strategies...")

86-87 – Question (i) is a little unclear to me. How about something like "which are the major gradients in the diversity of mammal and bird ecological strategies?". Whether you make this change is totally up to you.

Results

185-185 – I am not sold on the evidence presented for a "greater adaptive potential" of mammals compared to birds. Despite the greater variation in traits, adaptive potential will very much depend on the nature and type of threat (e.g. invasive species, climate change), such that it is hard to make any true general conclusions about overall adaptive potential. Perhaps put forward the greater adaptive potential as a hypothesis, stating that, however, it will depend upon the specific threat or sets of threats?

212-214 – It may be helpful to the reader to express contractions in niche space as percentages instead of unit-less n-dimensional volumes.

Discussion

295 – Correct to “body mass-associated extinction”

304 – Correct to “a species’ habitat use”

329-331 – I think there is a missed opportunity here to emphasize the importance of considering ecological traits, not just extinction risk, when prioritizing species/areas for conservation and tying this manuscript with literature on surrogacy among dimensions of biodiversity in conservation planning (e.g. Pollock et al. 2017 *Nature* 546: 141–144; Rapacciuolo et al. 2019 *Nature Ecology and Evolution* 3: 53-61; Girardello et al. 2019 *Scientific Reports* 9: 5636); this is of course if the authors wish to do so.

Projected losses of global mammal and bird ecological strategies

Reviewers' comments (black text)

Author responses (blue text)

REVIEWERS' COMMENTS:

Reviewer #2 (Remarks to the Author):

I have now reviewed the revised version of the manuscript “Projected losses of global mammal and bird ecological strategies”, paying particular attention to the point-by-point response to the comments raised by both reviewers on the previous version. I commend the authors on a truly great effort revising this manuscript – I think it is an improvement on an already excellent manuscript and it will provide an extremely useful contribution to the field.

Thank you very much for these comments! We really appreciate your positive feedback! Also thank you for taking the time to go through the point-by-point response to the comments raised by both reviewers on the previous version.

I only have very minor edits and a couple of suggestions (see below) – I will leave it to the editor and authors to decide whether these might be useful or not.

Finally, I was wondering if the authors were planning on releasing data on their estimated ecological trait spaces (e.g. species-level values for Fig. 1’s PC1 and PC2)? I am unsure whether this is a requirement or not in Nature Communications; however, I thought I would mention that I believe it would provide a fantastic resource for the research community and would undoubtedly stimulate many follow-up studies.

The species scores for the principal components are available on Github (https://github.com/03rcooke/hyper_pca) but have also now been added as a Source Data file following the Nature Communications guidelines. We hope they will prove useful and can contribute to follow-up studies!

Best,

Giovanni Rapacciuolo

Line edits (lines)

Abstract

27 – Suggest adding “future” after “projected”

Thank you, we have amended this in the manuscript.

Introduction

51 – correct to “a species’ ecological role”

Thanks! We hadn’t spotted this one, now corrected.

62-68 – I suggest moving the sentence with the prediction (“We predict that mammals will show...”) after that with the question (“how are ecological strategies...”)

Good point, that makes more logical sense! We have now changed this in the manuscript.

86-87 – Question (i) is a little unclear to me. How about something like “which are the major gradients in the diversity of mammal and bird ecological strategies?”. Whether you make this change is totally up to you.

Agreed, not the easiest to understand, we have altered to: “what are the major gradients across the diversity of mammal and bird ecological strategies?”

Results

185-185 – I am not sold on the evidence presented for a “greater adaptive potential” of mammals compared to birds. Despite the greater variation in traits, adaptive potential will very much depend on the nature and type of threat (e.g. invasive species, climate change), such that it is hard to make any true general conclusions about overall adaptive potential. Perhaps put forward the greater adaptive potential as a hypothesis, stating that, however, it will depend upon the specific threat or sets of threats?

Good point, we agree that our writing here was too definitive, so we have edited the text to read: “Mammals therefore show a greater range of ecological modes, which we hypothesize indicates both greater net evolutionary change - the dissimilarity between species regardless of the evolutionary pathways - and possibly greater ecological adaptive potential, which should enhance the probability that at least some species will survive into the future⁴⁴. Although, the adaptive potential of mammals and birds will depend on the specific nature and types of selection pressures.”

We chose to use the terminology ‘selection pressures’ instead of threats to reflect the context of the introduction.

212-214 – It may be helpful to the reader to express contractions in niche space as percentages instead of unit-less n-dimensional volumes.

We had the same thought, that it might be useful for the reader to include percentage reductions to express contractions in ecological strategy space. However, we have decided to continue only presenting the effect size volume reductions in niche space, as we think that percentages would underemphasise the absolute loss of ecological diversity and could therefore distract from the message that the effect of the projected loss is double that of random loss. For instance, we could liken it to presenting the number of extinctions as a percentage of the total, e.g., if 10% of species went extinct it would seem negligible but for mammals and birds that could mean the loss of over 1,500 species.

Discussion

295 – Correct to “body mass-associated extinction”

We have made this correction.

304 – Correct to “a species’ habitat use”

We have amended this in the updated manuscript.

329-331 – I think there is a missed opportunity here to emphasize the importance of considering ecological traits, not just extinction risk, when prioritizing species/areas for conservation and tying this manuscript with literature on surrogacy among dimensions of biodiversity in conservation planning (e.g. Pollock et al. 2017 Nature 546: 141–144; Rapacciuolo et al. 2019 Nature Ecology and Evolution 3: 53-61; Girardello et al. 2019 Scientific Reports 9: 5636); this is of course if the authors wish to do so.

Agreed! The multidimensionality of biodiversity is critical to better prioritise species conservation and understand biodiversity loss. The final paragraph now reads: “Here we highlight that continuing to protect the most at risk species could help to preserve a diversity of ecological strategies, which could be important for coping with environmental change⁵¹, and maintaining ecosystem functionality. Moreover, we suggest that greater consideration of the ecological importance and diversity of mammals and birds could benefit conservation planning. Our work therefore underlines the multidimensionality of biodiversity and suggests that analyses of conservation prioritisation across dimensions could be increasingly important into the future^{15,60,61}. Finally, forecasting the loss of ecological diversity and the associated functional consequences should improve our ability to predict and mitigate future responses that sustain ecosystems in the long-term.”